# Investigating the Relationship between Topographic Factors and Vegetation Spatial Patterns in the Alpine Plateau: A Case Study in the Southwestern Tibetan Plateau

**Yan Li** [1,2] , **Jie Gong** [1,]*  and **Yunxia Zhang** [1]

1  Key Laboratory of Western China's Environmental Systems, Ministry of Education (MOE), College of Earth and Environmental Sciences, Lanzhou University, Lanzhou 730000, China
2  Key Laboratory of Remote Sensing of Gansu Province, Heihe Remote Sensing Experimental Research Station, Northwest Institute of Eco–Environment and Resources, Chinese Academy of Sciences, Lanzhou 730000, China
*  Correspondence: jgong@lzu.edu.cn; Tel.: +86-138-9325-6119

**Abstract:** Vegetation on the Southwestern Tibetan Plateau (SWTP) is critical to ensuring ecological security and promoting regional economic and social development. Here, we explored the relationship between topographic factors (elevation, slope, and aspect) and the spatial patterns in the normalized difference vegetation index (NDVI) in the SWTP over the past 20 years. The results found that the NDVI in the SWTP was primarily influenced by elevation and slope. The regions with significant variations in NDVI were concentrated between 4500 m to 5500 m and slopes ranging from 0° to 15°. Although the influence of aspect on NDVI was small, there was a decreasing trend in NDVI on sunny slopes and an increasing trend on shady slopes. Dominant topographic conditions were identified by considering 230 different combinations of elevation, slope, and aspect. The combination of topographic parameters indicated stronger patterns in NDVI variability, notably within sections of 0°–25°slopes and below 5000 m elevation. These findings highlight the relevance of topography, notably slope and aspect, for vegetation in alpine settings. The information gathered from this study about the prevailing topographic distribution and vegetation growth state in the SWTP can help with future ecological restoration and conservation efforts in the Tibetan Plateau and other comparable regions worldwide.

**Keywords:** NDVI; spatiotemporal change; Tibetan Plateau; terrain factors; the distribution index

## 1. Introduction

Vegetation is a crucial component of the Earth's carbon cycle and an important indicator of the health of terrestrial ecosystems [1,2]. Vegetation change influences not only climate regulation, hydrological processes, and ecosystem stability but also serves as an indicator of an ecosystem's adaptation to climate change, natural environments, and human activities [3,4]. The overall environmental condition of a region can be assessed by analyzing the spatial distribution pattern of vegetation, its dynamic, and its response to influencing factors [5,6]. This has become a focus for scholars and governments [7,8].

Remote sensing technology and methods have rapidly developed to monitor and evaluate vegetation status on a large spatial scale and over time [9–11], has and have provided a wealth of information and new technological support, especially in mountainous areas with limited transportation and field observation [10,12]. The normalized difference vegetation index (NDVI), with values ranging from −1 to 1, is widely used in monitoring vegetation cover, vegetation growth status, environmental changes, and ecosystem change [11,13]. Generally, the areas with an NDVI value below 0.1 are considered non-vegetated regions, such as snow-covered regions, water bodies, or bare land. Conversely, the areas with NDVI

values above 0.1 are considered vegetated regions, indicating better vegetation growth and denser coverage as the value increases [14,15].

NDVI data and analysis methods, such as trend analysis and correlation analysis, are often used to explore the changing trends of NDVI and their relationship and influencing factors to reveal the impacts of climate change and human activities on vegetation [9–13]. Some scholars also apply models such as geographically weighted regression to quantify the main influencing factors of NDVI change [14]. However, with the intensification of climate warming and rapid socioeconomic development, more studies are focused on climate change and human activities, often overlooking the role of topographical factors in NDVI [15–17]. The process of vegetation growth is related to climate change and influenced by the terrain in which it is located [18]. Terrain affects vegetation's adaptability, growth status, and distribution patterns by controlling the spatial distribution of environmental factors, such as solar radiation, soil type, and soil moisture, leading to spatial heterogeneity and distinct vertical zonation of vegetation [19–21]. Therefore, terrain can indirectly affect vegetation's distribution patterns and changing trends [22]. Especially in mountainous ecosystems, vegetation distribution and changing trends are more complex [23–25]. Thus, more attention is needed to explore how terrain factors individually and collectively influence the growth status and distribution patterns of the NDVI, which can contribute to understanding the distribution patterns and future evolution trends of regional vegetation with a significant practical application.

The Tibetan Plateau (TP) is a mountainous ecosystem offering much biogeographical information in the vertical direction [26]. The spatial distribution of its ecosystems is significantly influenced by topography [27]. Additionally, diverse climatic conditions and complex topographic factors, such as elevation and slope, influence the vegetation, soil, and entire natural complex ecosystems of the TP [28,29]. These topographic factors alter the water and thermal balance and modify the soil's physical and chemical conditions, resulting in the vertical zonation of vegetation [22]. However, recent studies on the dynamics of vegetation changes in the TP have primarily focused on climate change and human activities, often overlooking the relationship between topography and vegetation [30,31]. Although some studies have shown that topographic factors influence vegetation, the effect of elevation on vegetation is more significant than that of slope and aspect. This result often overlooks the effects of slope and aspect on vegetation [32,33]. Slope reflects the degree of surface inclination, impacting energy conversion and the flow of materials on the Earth's surface. At the same time, this aspect impacts the spatial distribution of solar radiation and soil moisture [34,35]. Under similar elevation conditions, slope and aspect directly influence the spatial redistribution of solar radiation and precipitation, creating local micro-climates that subsequently impact the distribution patterns of plants [36,37]. Therefore, it is crucial to understand the comprehensive impact of elevation, slope, and aspect on vegetation distribution patterns and their variation [38]. The relationship between vegetation change trends and topographic factors can aid ecological restoration and management, especially in fragile alpine mountainous areas.

The Southwestern Tibetan Plateau (SWTP), located in the southwestern part of the TP, belongs to a typical alpine mountainous ecosystem surrounded by the Himalayas, the Gangdis–Nyenching Tanggula Mountains, and the Karakoram Mountains [39]. The entire region boasts a myriad of complex and diverse landforms, with vast mountain ranges and breathtaking landscapes of towering peaks, deep valleys, intersecting ravines, and meandering river valleys [40]. The intricate topography and varying elevations shape the climatic conditions of the plateau, leading to a nuanced distribution of vegetation and distinct vertical zonation [41]. The local vegetation types are alpine grasslands, alpine meadows, and desert grasslands, which make it a typical representative region of agriculture and animal husbandry on the TP [39,40,42]. Based on fieldwork and statistical data, there was a consistent increase in livestock populations before 2010a, resulting in the overexploitation of grassland resources and the degradation of grassland ecosystems in certain regions [43]. Various ecological and environmental projects have been implemented

on the TP since 2010 [42]. Implementing these ecological and environmental projects has led to an overall improvement of the regional environmental quality. However, it is essential to note that certain local areas have experienced degradation despite these efforts [42,44]. Considering the vital role of the local grassland ecosystem in ecological functions and service, as well as the harsh climatic conditions, the degradation of grassland resources would directly reduce the productivity and resilience of the ecosystem. Simultaneously, it would introduce new constraints to grass and livestock production, livelihoods, and even the relationship between humans and nature, challenging vegetation restoration efforts. Therefore, it is crucial to understand the spatial distribution characteristics of vegetation and their relationship with topographic factors in the SWTP to gain deeper insights into the patterns, mechanisms, and interactions of vegetation growth. This will contribute to an understanding of regional vegetation distribution patterns and to future change and practical implications.

Using MODIS NDVI data from 2000 to 2020, combined with trend analysis and terrain distribution index methods of topographic data (elevation, slope, and aspect), we explored the spatiotemporal dynamics of vegetation and their relationship with topographic factors in the SWTP over the past two decades. We aimed to: (1) analyze the spatiotemporal variations of NDVI; (2) explore the relationships between NDVI and individual topographic factors (elevation, slope, and aspect); (3) reveal the distribution characteristics of NDVI change types under the comprehensive impact of topographic factors and discover the range of areas sensitive to vegetation changes. The results are expected to enhance the understanding of the NDVI changing characteristics under different topographies, and to tailor corresponding adaptation strategies to protect fragile alpine grassland ecosystems. Furthermore, they will provide a scientific basis for alpine ecological management and sustainable development on the TP [45].

## 2. Material and Methods

### 2.1. Case Study Area

The SWTP spans a vast area of 308,900 km$^2$ and is situated geographically between 26°59′–33°97′N and 78°28′–90°40′E [46]. The SWTP is located at the convergence of the Karakoram Mountains, the Himalayas, the Ganges, and other mountain ranges. It is the source of numerous international rivers, including the Yarlung Zangbo River, the Indus, and the Ganges. As "the cradle of many mountains and the origin of many rivers," it plays a crucial role as a regional ecological barrier [47]. The SWTP is renowned for its cold and arid climate, with an average elevation exceeding 4000 m. The primary vegetation types are alpine meadows and desert grasslands. The SWTP covers two prefectures of Ngari and Rikaze and included 24 counties and districts (Figure 1). The population is relatively sparse, with the main concentration in the city of Rikaze in the SWTP. Human activities primarily occur in valley areas that offer flat terrain, convenient transportation, and favorable natural conditions, including Zhongba, Gê'gya, Ngamring, and Xietongmen Counties. However, the SWTP, characterized by fragile environmental conditions, is prone to frequent natural disasters, such as heavy rain, snowfall, hailstorms, landslides, mudslides, and the freezing and thawing of permafrost.

### 2.2. Data

#### 2.2.1. NDVI Data

NDVI data from the MODIS sensors provide reliable information on the vegetation dynamics and changes over time. By analyzing MODIS-based NDVI data, researchers can gain insights into the ecological conditions and vegetation change trends in the TP [48]. The NDVI data used in this study are from the Terra MODIS dataset (MOD13Q1 product) from 2000–2020, with a temporal resolution of 16 days and a spatial resolution of 250 m. We preprocessed the NDVI data using the maximum value composite (MVC) [49] to eliminate the impacts of clouds, aerosols, and the solar zenith angle to obtain an annual average NDVI dataset for the last 20 years via the Google Earth Engine (https://earthengine.google.com/,

accessed on 10 December 2021) [50]. Due to the sparse vegetation cover in some subareas of the SWTP, and after considering the relevant literature, this study ultimately excluded pixels with NDVI values below 0.1 for a continuous period of 15 years. The remaining areas were used as the primary study regions in this research [51,52].

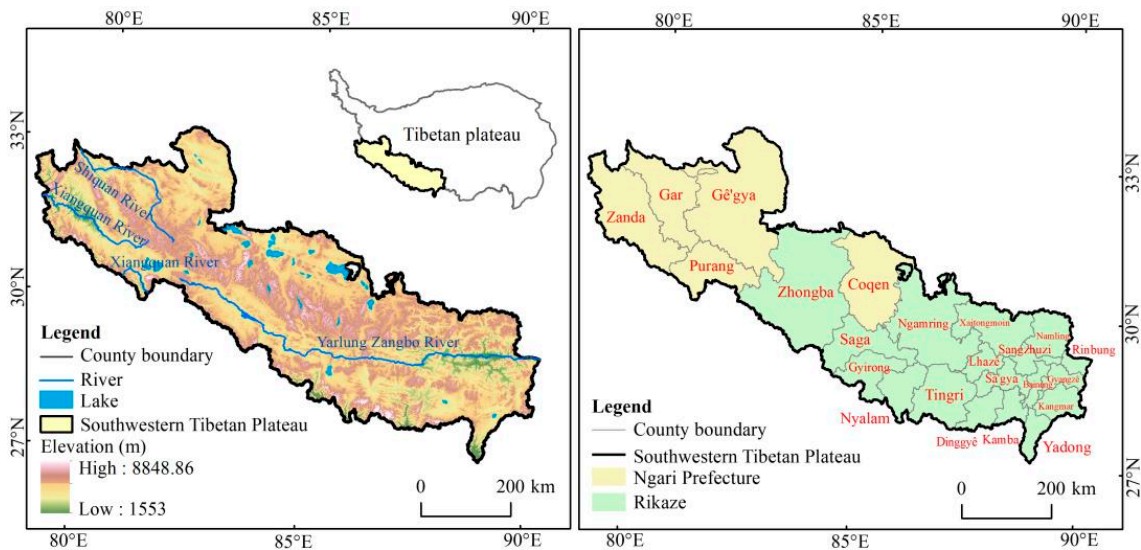

**Figure 1.** Study area of the SWTP.

### 2.2.2. Terrain Data

The Geospatial Data Cloud Platform (https://www.gscloud.cn/, accessed on 10 December 2021) provided the ASTER GDEM elevation data with a spatial resolution of 30 m. To create topographic factors maps of elevation, slope, and aspect, we used ArcGIS software 10.8.1 to perform resampling (Figure 2). The elevation of the SWTP is in the range of 1553–8846.86 m, and was grouped into five classes at 500 m increments: below 4000 m, 4000–4500 m, 4500–5000 m, 5000–5500 m, and above 5500 m. Following the classification criteria of geomorphological maps established by the International Geographical Union Commission on Geomorphological Surveys and Geomorphological Mapping, the slope was categorized into five classes: 0°–5°, 5°–15°, 15°–25°, 25°–35°, and greater than 35° [53]. The slope orientation was also divided into five classes: flat (−1°), shady slope (0°–67.5°, 337.5°–360°), sunny slope (157.5°–247.5°), half shady slope (67.5°–112.5°, 292.5°–337.5°), and half sunny slope (112.5°–157.5°, 247.5°–292.5°) [54]. Figures 2 and 3 display the statistical results depicting the spatial distribution and topographic differentiation of all topographic factors.

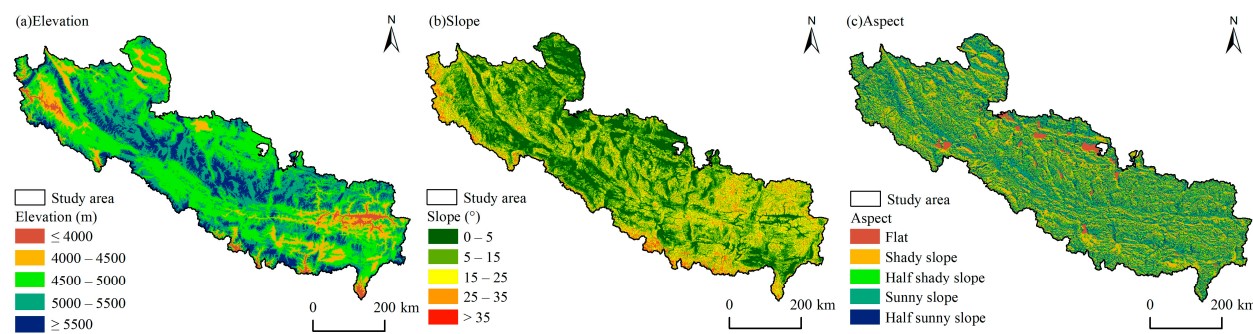

**Figure 2.** Terrain factors of the study area of the SWTP.

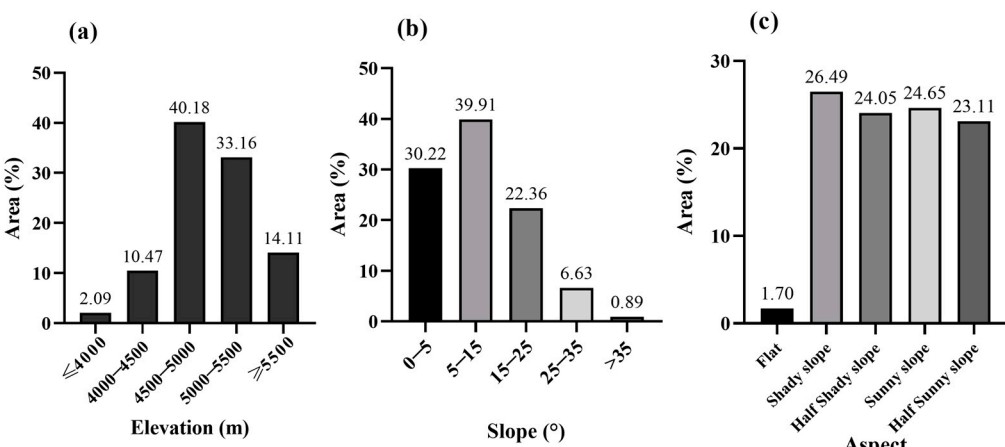

**Figure 3.** Classification and area of topographic factors in the study area. (**a**) Elevation; (**b**) Slope; (**c**) Aspect.

### 2.3. Methods

#### 2.3.1. Trend Analysis

This study calculated the annual average NDVI from 2000 to 2020 and explored the spatiotemporal variation of NDVI on the SWTP using linear regression analysis [55]. This method enables the simulation of NDVI on a pixel-by-pixel basis while mitigating the impact of extreme data [56]. The calculation formula is:

$$\theta_{slope} = \frac{n \times \sum_{i=1}^{n} i \times \overline{X}_i - \sum_{i=1}^{n} i \sum_{i=1}^{n} \overline{X}_i}{n \times \sum_{i=1}^{n} i^2 - \left(\sum_{i=1}^{n} i\right)^2} \tag{1}$$

where $\theta_{slope}$ indicates the trend of vegetation dynamics during the study period. Positive and negative slopes indicate positive and negative trends in vegetation dynamics; $n$ is the length of the research period (in this case, $n = 21$). The annual average NDVI in year $i$ is $\overline{X}_i$. The F test was used to determine the significance level of vegetation change. The trend change can be classified into five classes: no significant differences ($\theta = 0$), significant increase ($\theta > 0$, $p \leq 0.05$), no significant increase ($\theta > 0$, $p > 0.05$), significant decrease ($\theta < 0$, $p \leq 0.05$), no significant decrease ($\theta < 0$, $p > 0.05$).

#### 2.3.2. Terrain Distribution Index

Terrain factors are among the natural factors that influence the spatiotemporal patterns of NDVI [57]. The influence on NDVI can be simplified as the frequency of occurrence of NDVI and its change types along the terrain gradient. This study calculated the terrain distribution index, a dimensionless index, to analyze the distribution characteristics of NDVI and its change types across different terrain gradients. This index calculates the ratio of the probability of NDVI (or NDVI change types) occurring under a specific terrain condition to the probability of that terrain condition occurring in the study area, revealing the distribution changes of NDVI (or NDVI change types) on terrain levels. It shows the spatial quantitative relationship between NDVI and three terrain factors: elevation, slope, and aspect [58,59]. The calculation formula is:

$$P = \frac{(S_{ie}/S_i)}{(S_e/S)} = \frac{S_{ie}}{S_e} \times \frac{S_i}{S} \tag{2}$$

where $P$ is the terrain distribution index, $S_{ie}$ is the $i$th NDVI (or NDVI change type) area under $e$ topographic conditions, $S_i$ is the total area of the $i$th NDVI (or NDVI change type); $S_e$ is the total area of the $e$th terrain; $S$ is the total area of the survey region. When $P > 1$, it indicates that the terrain $e$ is the dominant terrain of the NDVI (or NDVI change types), and the higher of $P$ value, the more significance in the degree of dominance. When $P = 1$, the $i$th NDVI (or NDVI change types) is more stable under the $e$ terrain. When $P < 1$, it indicates that the $e$ terrain is not the main distribution terrain for the $i$th NDVI change type.

## 3. Results Analysis

### 3.1. Spatiotemporal Variation Characteristics of the NDVI from 2000 to 2020

Over the past 20 years, the NDVI in the SWTP has shown an overall increasing trend with some local decreases. From 2000 to 2020, the NDVI change rate ranged from −0.019/a to 0.030/a in the SWTP. Overall, 75.58% of the SWTP exhibited an increasing trend in NDVI, while 24.32% showed a decreasing trend, indicating significant spatial heterogeneity over the past two decades (Figure 4). Significant changes in NDVI were experienced by 25.74% of the SWTP (Figure 4b). The subregions with significant increases accounted for 23.74% and were mainly distributed in the southwest and central–eastern SWTP. On the other hand, the subregions with significant decreases accounted for 2.00% and were concentrated in Zhongba County and the southeastern areas of Dingjie County and Gangba County (Figure 4b).

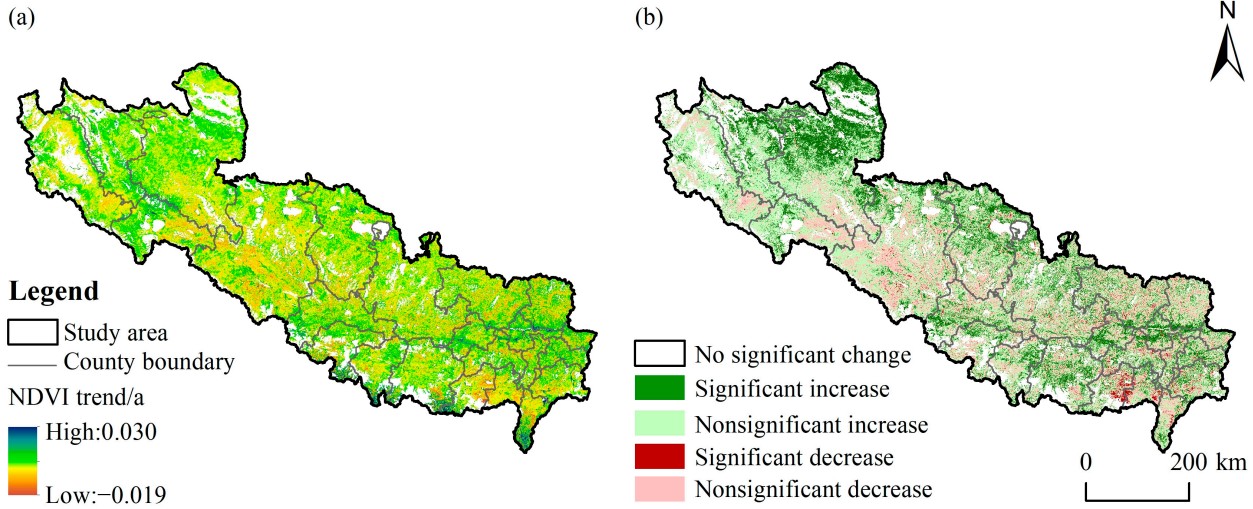

**Figure 4.** Annual trends of NDVI (**a**) and significance test results (**b**) on the SWTP from 2000 to 2020.

### 3.2. Impact of Terrain Factors on NDVI

#### 3.2.1. Impact of Elevation on NDVI

The average NDVI value in the SWTP exhibited a decreasing trend with elevation increasing. The subregions with significant NDVI changes were mainly concentrated in the 4000–5500 m range (Figure 5). The significant increases primarily occurred at 4500–5000 m and 5000–5500 m. The range of 4500–5000 m was the primary area with a significant increase in NDVI at 47.28% of the significantly increased area. Significant decreases mainly occurred in the 4000–5000 m elevation range, primarily in the 4500–5000 m range, concerning 52.33% of the significant decrease area. The regions with no significant decrease were distributed within the 4500–5500 m range; 4500–5000 m was the primary area with no significant decrease, representing 43.47% of the regions with no significant decrease.

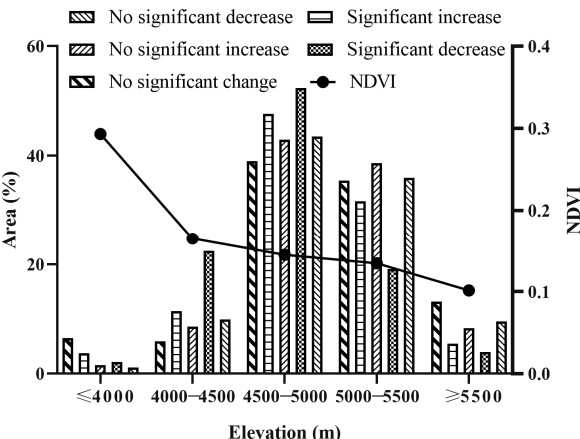

**Figure 5.** Percentage of NDVI values and NDVI change types in different elevations.

### 3.2.2. Impact of Slope on NDVI

The average NDVI in the SWTP increased with slope. The vegetation change types with significant changes were concentrated in two slope ranges, 0°–5° and 5°–15° (Figure 6). Among these, the subarea with the most significant NDVI decreases were in the 0°–5° range, accounting for 42.83% of the substantial reduction area. The main subarea with no significant decrease in NDVI was in the 5°–15° range, accounting for 42.91% of the subareas with no significant decrease. Furthermore, the 5°–15° range was the main subarea with both significant-increase areas and no-significant-increase areas in NDVI, accounting for 40.21% and 43.61% of the significant-increase and no-significant-increase areas, respectively.

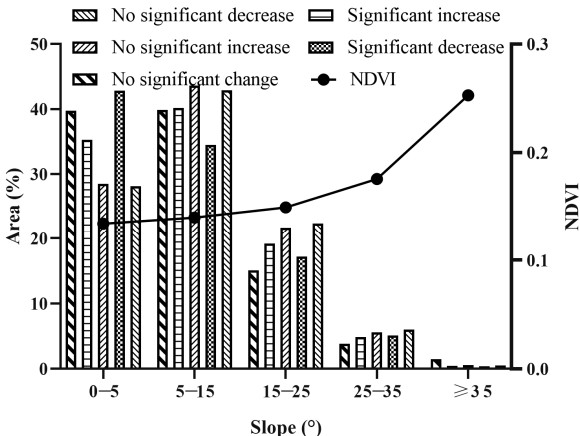

**Figure 6.** Percentage of NDVI values and NDVI change types in different slopes.

### 3.2.3. Impact of Aspect on NDVI

There were no significant differences in the average NDVI and change types induced by aspect in the SWTP (Figure 7). However, the NDVI values of shady slopes were generally higher than those of sunny slopes. As to the areas of significant decrease and no significant decrease, the area distribution on sunny slopes was higher than on shady slopes. On the other hand, areas with significant increases or no significant increases were more common on shady slopes. Sunny slopes had the highest concentration of NDVI change types with significant and insignificant decreases, accounting for 29.46% and 27.32% of the total decrease areas, respectively. On the other hand, shady slopes had the highest concentration of NDVI change types of significant and insignificant increase, accounting for 27.74% and 26.31% of all increased areas, respectively.

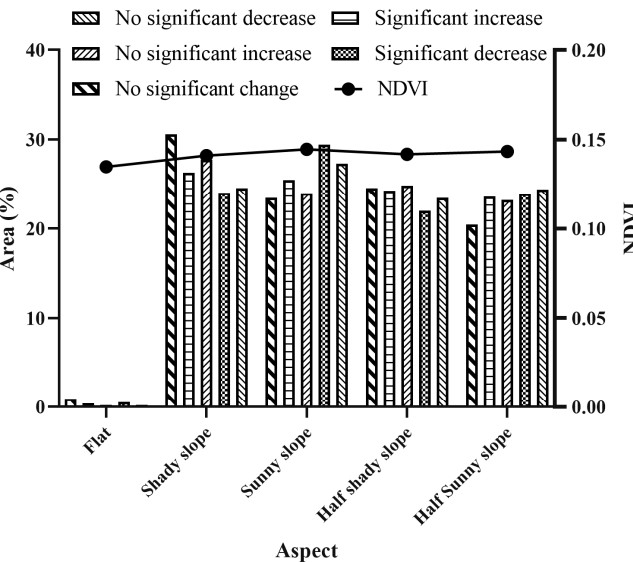

**Figure 7.** Percentage of NDVI values and NDVI change types in different aspects.

### 3.3. Impact of a Single Topographic Factor on the NDVI Change Types

#### 3.3.1. Impact of Elevation on the NDVI Change Types

The distribution index of NDVI change types in the SWTP exhibited distinct patterns depending on elevation (Figure 8). The type with a significant decrease was dominant below 5000 m ($1.071 < P < 2.284$), with the most dominant terrain at 4000–4500 m ($P = 2.284$). The no-significant-decrease type was dominant above 5500 m ($P = 1.218$), and in the altitude range of 4000–5500 m, the no-significant-decrease type remained stable. The type with a significant increase was dominant below 5000 m ($1.074 < P < 1.892$), with the most dominant terrain below 4000 m ($P = 1.892$). The type with no significant increase was dominant above 5000 m ($1.061 < P < 1.074$).

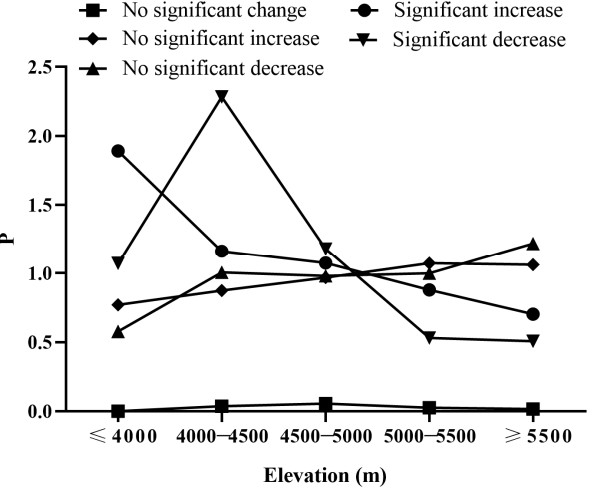

**Figure 8.** NDVI change types at different elevation grades in the SWTP.

#### 3.3.2. Impact of Slope on the NDVI Change Types

The distribution index of NDVI change types on the SWTP did not exhibit significant differentiation (Figure 9). The type with significant decrease was dominant on 0°–5° ($P = 1.412$), and as the slope increased, the P value fluctuated within the range of 0.742 to 0.927. The type with no significant decrease became dominant after the slope exceeded 5° ($1.096 < P < 1.011$) and remained stable within this range. The type with significant increase was dominant on flat slopes ($P = 1.164$), and as the slope increased, the *p* value of this type

fluctuated within the range of 0.856 to 0.947. The type with no significant increase became dominant within the range of slopes greater than 5° (1.014 < P < 1.074).

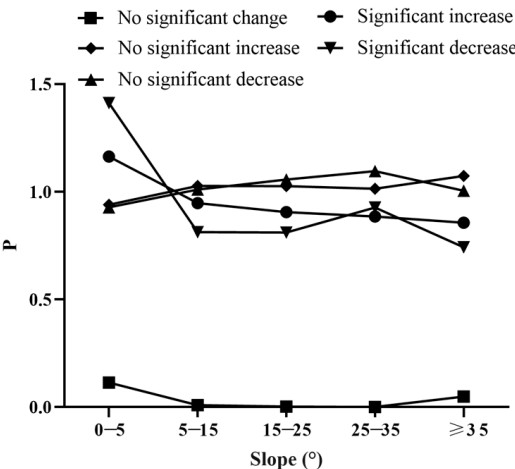

**Figure 9.** NDVI change types at different slope gradients in the SWTP.

### 3.3.3. Impact of Aspect on the NDVI Change Types

The distribution index of the NDVI change types exhibited a relatively gradual variation with aspects in the SWTP (Figure 10). The significant decrease in NDVI primarily occurred in the subareas with flat, sunny, and half-sunny slopes. The corresponding *p* values for these terrains were 2.004, 1.169, and 1.016, respectively, with the flat slope being the most dominant terrain of NDVI change. The no-significant-decrease type dominated on the sunny and half-sunny slopes, with *p* values of 1.084 and 1.035, respectively. The considerable increase type prevailed on the flat slope, sunny slope, and half-sunny slope, with *p* values of 1.384, 1.011, and 1.000, respectively. The flat slope exhibited the highest dominance for this NDVI change type among these terrains. The no-significant-increase type prevailed on the shady and half-shady slopes, with *p* values of 1.042 and 1.021, respectively. The flat slope exhibited the highest dominance for this NDVI change type among these terrains. The no-significant-increase type prevailed on the shady and half-shady slopes, with *p* values of 1.042 and 1.021, respectively.

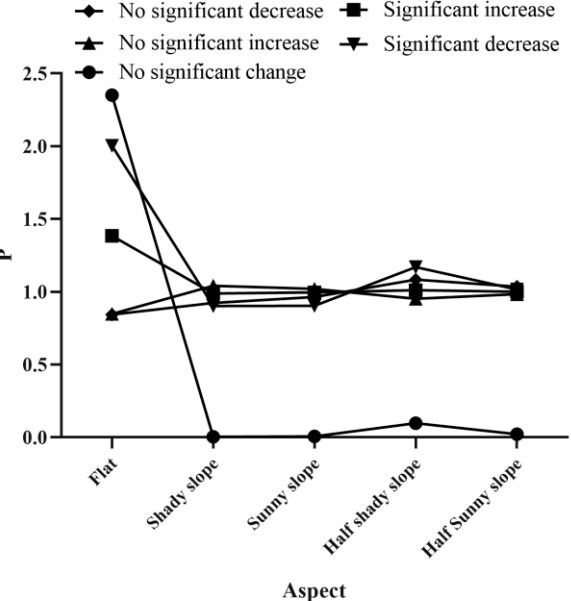

**Figure 10.** NDVI change types at different aspects in the SWTP.

*3.4. Synergistic Impact of Topographic Factors on the NDVI and NDVI Change Types*

3.4.1. Synergistic Impacts of Topographic Factors on the NDVI

Low-cover vegetation (NDVI ≤ 0.2) exhibited a gradual increase in the distribution index for each terrain combination as the elevation increased. Medium- to high-cover vegetation (0.2 < NDVI < 0.4) initially showed an increase and then a decrease in the distribution index for each terrain combination as elevation increased, while high-cover vegetation (NDVI ≥ 0.4) demonstrated a gradual decrease (Figure 11). At altitudes lower than 4000 m, slopes ranging from 0° to 15° and flat terrains greater than 35° were the dominant terrain types (0.2 < NDVI < 0.4). At an NDVI greater than or equal to 0.4, all terrains (excluding flat slopes) from 5°–15° and greater than 35° were dominant (Figure 11a). For elevations between 4000 to 4500 m, all terrains with slopes greater than 15° (0.2 < NDVI < 0.4) were dominant, while all terrains with slopes greater than 5° were also dominant (NDVI ≥ 0.4) (Figure 11b). In the range of 4500 to 5000 m, areas with slopes greater than 5° were dominantly distributed (0.2 < NDVI < 0.4), whereas sunny slopes with angles greater than 35° were the dominant terrain (NDVI ≥ 0.4) (Figure 11c). In the range of 5000 and 5500 m, sunny slopes ranging from 25°–35° were dominant (0.2 < NDVI < 0.4) (Figure 11d). At elevations higher than 5500 m, the *p*-values of all terrain were more significant than 1, and low-cover vegetation had a dominant distribution of terrain at higher elevations (0.2 < NDVI < 0.4) (Figure 11e).

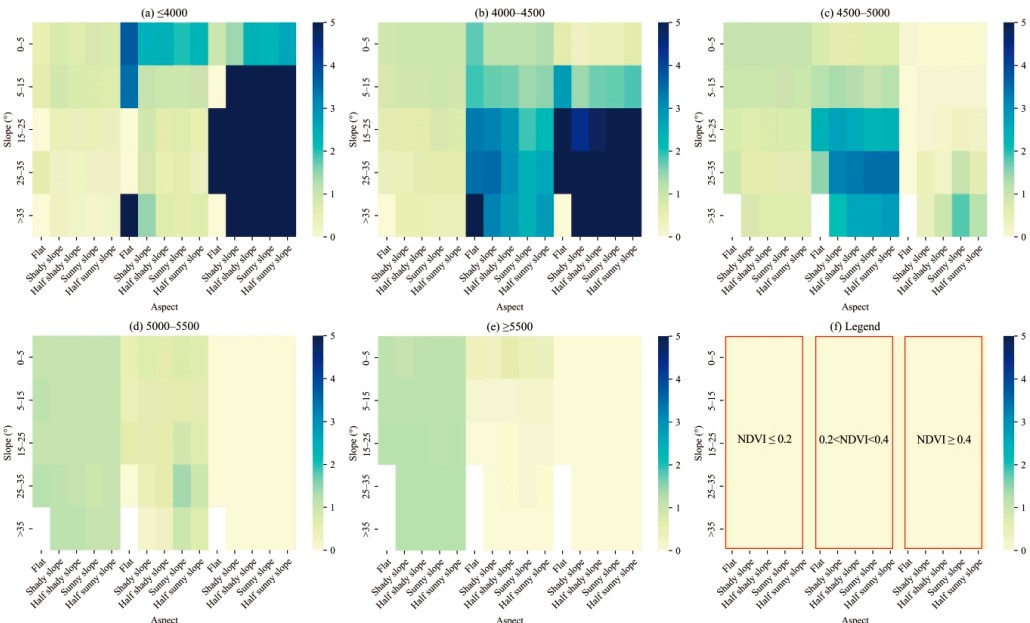

**Figure 11.** Distribution index of NDVI over terrain combinations in the SWTP (Note: The horizontal axis represents the aspect, with separate sections for NDVI ≤ 0.2, 0.2 < NDVI < 0.4, and NDVI ≥ 0.4. The right Y-axis indicates the magnitude of the distribution index of NDVI under specific terrain combinations, with the color scale representing its value. Similar values within a range of elevation are assigned the same color, indicating similar vegetation growth patterns in those terrains. A higher *p*-value suggests a more significant distribution advantage for vegetation under that terrain combination, while a lower *p*-value implies less favorable conditions for vegetation growth. If the color is white, such a terrain combination does not exist).

3.4.2. Synergistic Impacts of Topographic Factors on the NDVI Change Types

NDVI types with significant changes were mainly concentrated in the subareas below 4500 m. In contrast, NDVI types with insignificant changes tended to have a distribution index closer to 1 with the elevation increased (Figure 12). The kinds of NDVI changes varied considerably across the distribution range, with 230 out of 363 combinations of elevation, slope, and aspect terrain being dominant in the SWTP. The significant NDVI

increase type prevailed at elevations lower than 4000 m, the slope in the 0°–15° and each aspect (Figure 12a). The significant-decrease NDVI type had a concentrated distribution in the 4000–4500 m range, the slope in the range 0°–5°, and each slope (Figure 12b). The no-significant-NDVI-increase type was more common at elevations lower than 4500 m, with a slope greater than 15° of each aspect (Figure 12a,b). At elevations higher than 4500 m, the no-significant-NDVI-increase type and the distribution index gradually increased (Figure 12c–e). The no-significant-NDVI decrease type had no distribution advantage at elevations lower than 4000 m (Figure 12a). When the elevation gradually increased, the *p* value tended to be close to 1 on the sunny and half-sunny slopes and progressively gained a distribution advantage (Figure 12b–e).

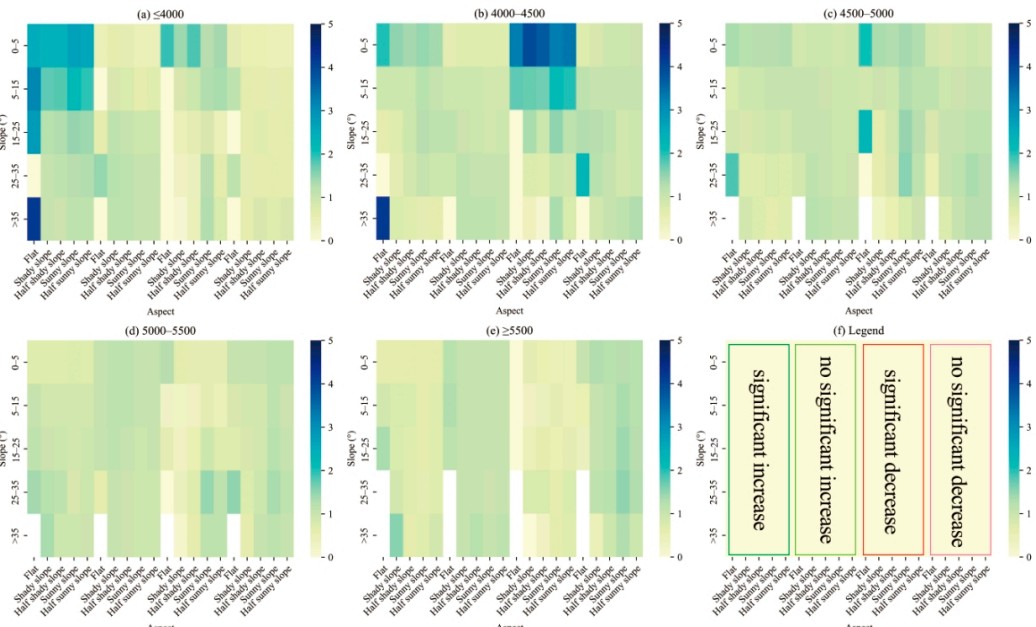

**Figure 12.** Distribution index of NDVI change types over terrain combinations in the SWTP. (Note: The horizontal axis represents the aspect, with separate sections for NDVI change types. Other information is the same as above).

## 4. Discussion

### 4.1. Temporal and Spatial Distribution Characteristics of NDVI

This study utilized MODIS NDVI data and trend analysis to analyze the NDVI trend changes in the SWTP over the past 20 years. There is an overall increasing trend in NDVI with some localized decreasing trends. This finding is consistent with previous studies on the TP [60,61]. Additionally, we found NDVI increasing mainly in the southwestern and east–central regions, where the terrain is relatively gentle. This may be attributed to a series of conservation policies, such as the conversion of farmland to forests and grassland, the conversion of grazing land to grassland, and grassland ecological conservation projects, which contributed to the vegetation cover improvements [58]. The NDVI mainly decreased in Zhongba County, a major pastoral county in the SWTP with large livestock populations. Despite the implementation of the environmental conservation program, which has reduced grazing intensity in Zhongba County, a large livestock population and overgrazing have led to NDVI destruction in the SWTP [40]. Thus, while the environmental conservation program has improved NDVI on most of the SWTP, parts with extensive animal husbandry and long-term grazing activities decreased NDVI. Therefore, long-term and locally appropriate ecological restoration programs are needed.

### 4.2. Relationship between a Single Topographic Factor and NDVI

This study examined the correlation between different topographic factors and the trends of NDVI and NDVI change types via the terrain distribution index. The influence of topographic factors on NDVI can be ranked as elevation, slope, and aspect, which is consistent with the findings of previous studies [18,32,62]. In the SWTP, NDVI shows significant increases in the subareas below 4000 m. At the same time, vegetation degradation is more apparent in the 4000–5000 m range (Figures 5 and 8), such as in the Sangzhuzi District of Rikaze City, with an elevation primarily below 4000 m and a large amount of arable land. The local government emphasizes environmental protection when expanding towns and cities by implementing tree planting and grass seeding [18]. As to the subareas with elevation above 5000 m, alpine climatic conditions and the human living environment have shifted from agriculture to a combination of cropping and livestock husbandry. This transition has increased the dependence on natural pasture in the river valleys [39]. We further investigated the relationship between slope and NDVI change, finding a noteworthy reduction in vegetation in areas with gentle slopes ranging from 0°–5°. On the other hand, the subregions with slopes ranging from 5°–15° experienced a substantial NDVI increase after an initial decline. This finding is consistent with the fact that human activities are more concentrated in subareas with gentle slopes. It is important to note that although the influence of aspect on NDVI is relatively minor, there is still a discernible pattern. Specifically, NDVI on sunny slopes exhibits a higher frequency of significant decreases, whereas NDVI on shady slopes shows a slightly higher occurrence of significant increases. To investigate this phenomenon and account for the impact of other factors, such as climate, grazing, and slope, further analyses can be performed in the subareas with similar slopes and interannual grazing intensities. Furthermore, utilizing long-term vegetation sampling sites in homogeneous regions can help to better understand the connection between aspect and pattern and trend of NDVI distribution.

### 4.3. Relationship between Multiple Topographic Factors and NDVI

Vegetation growth is closely related to geographic factors. The influence of terrain on vegetation growth is a complex and comprehensive process involving multiple driving factors [63]. In this study, we found that the dominant distribution of low vegetation cover (NDVI $\leq$ 0.2), moderate vegetation cover (0.2 < NDVI < 0.4), and high vegetation cover (NDVI $\geq$ 0.4) were concentrated at elevations greater than 5000 m, 4000–5000 m, and below 4500 m, respectively. While forests were mainly distributed in the subareas below 4500 m, grasslands and meadows were primarily distributed between 4000–5000 m, and desert grasslands were mainly distributed above 5000 m. This result is consistent with the characteristics of the altitudinal gradient of vegetation in the SWTP [41]. Additionally, we found that the apparent NDVI changes were mainly concentrated at elevations lower than 5000 m with slopes of less than 15°. This is because vegetation is more affected by human activities in areas below 5000 m due to the limitations of topographic factors. Above 5000 m, the vegetation type was mainly desert steppe, which was affected by climate change rather than topographic factors. Therefore, the distribution index of significant NDVI change types was significantly greater than 1 in the slope range of 0°–15° and below 4500 m, while the distribution index of no-significant-change types was close to 1 in all the terrain combinations at elevations greater than 5000 m. Zhang et al. (2019) found that vegetation changed with elevation, showing changes in sensitivity to human activities and climate change in the TP [38]. This demonstrates a significant relationship between topography and NDVI that influences vegetation's change and distribution pattern [64].

In the future, it will be necessary to pay more attention to the subregions with altitudes lower than 5000 m and slopes of 0°–25° and to propose scientific pasture management and effective governance. Rational land-use planning and urban planning are needed. In addition, using the topographic information of NDVI distribution, we can provide practical guidance for environmental protection and restoration and ecological engineering in

different subregions. It is crucial to adapt to local conditions to realize regional sustainable development [45].

*4.4. Limitations and Outlook*

This study aimed to uncover the relationship between NDVI and its changing types with terrain factors (elevation, slope, and aspect) in the SWTP. We found that two key area types were those with elevations lower than 5000 m and those with slopes of 0°–25°. These areas often experience frequent human activities, such as urbanization and grazing. Therefore, it is recommended to protect the primary environmental conditions and natural vegetation during urbanization and construction activities. It is vital to implement adaptive zoning protection strategies based on zoning of elevation, slope, and aspect. Moreover, it is essential to strengthen the environmental construction projects, such as artificial turf planting, reforestation (and/or grassland restoration), and conservative grazing. It is necessary to calculate suitable carrying capacity based on actual pasture productivity to balance foraging and livestock via proper management practices, such as conservative grazing and enclosures, and to prevent overgrazing. In addition, we found that the influence of slope and aspect on NDVI was more significant on a small scale. The NDVI on shady slopes primarily increased, and thus shady slopes can be prioritized for production functionality. Therefore, an alternating grazing/alternating functional zone approach can be implemented to ensure coordination between production functional areas and restoration functional areas. In summary, it is critical to consider local conditions and integrate the patterns of terrain and vegetation distribution to integrate ecological, economic, and social benefits through environmental conservation policies and human activities governance in the SWTP. This would be helpful for pasture conservation and livestock husbandry, environmental security, and sustainable development in the alpine plateau.

However, this study has some uncertainties and limitations. Firstly, this work divided elevation, slope, and aspect into five classes based on the rules of the International Geographical Union Commission on Geomorphological Surveys and Geomorphological Mapping. The classification and analysis may be coarse and result in a lack of distinct distribution characteristics of NDVI in each segment, thus potentially causing analytical errors. In the future, more efforts will be needed to refine the slope and aspect classes on more minor scales and explore how they impact vegetation at a given elevation. Secondly, in addition to the direct impact of topographic factors, terrain also influences vegetation pattern via regional climatic conditions and human activities. Currently, the relationship between topography and these factors and their joint influences on vegetation distribution remains relatively unknown. In the future, more field observation work and multiple remote sensing images are needed to study the relationship between vegetation cover and topographic factors, especially in the subareas of 4000–5000 m and 0°–25°. Biophysical models of soil–vegetation–hydrology relationships could be used to investigate the relationships and mechanistic linkage between ecological, hydrological, and soil processes [65].

Finally, the distribution index, obtained from the standardized dimensionless data, is helpful to avoid the influence of absolute area differences caused by classification or grading of the evaluation results. The results demonstrate that the distribution index effectively reveals vegetation distribution patterns under single and combined terrain gradients. Moreover, it provides new perspectives and research methods for studying the relationship between topographic factors, ecosystem services, and land use patterns. In the future, it will be essential to investigate how to combine the distribution index with ecosystem services and landscape patterns to explore the spatial relationships between topographic factors influencing landscape patterns and regional ecosystem services.

**5. Conclusions**

This paper evaluated the distribution pattern and change trend of NDVI in the SWTP from 2000 to 2020 via trend analysis and the terrain distribution index. The study also investigated the relationship between terrain gradient and NDVI and NDVI change types

using three terrain factors (elevation, slope, and aspect). In the past 20 years, the rate of change of NDVI fluctuated between −0.019 and 0.030 per year, with an overall increasing trend and some local decreasing trends. The correlation between elevation, slope, and NDVI values was more significant, while the correlation between NDVI and aspects was weaker. In general, NDVI values decreased with the increase in elevation and slope. The relationship between vegetation change types and elevation was mainly concentrated in the elevation range of 4500–5500 m. Similarly, the NDVI change types related to slope were most prominent at 0°–15°. The NDVI on shady slopes increased significantly, while NDVI on sunny slopes decreased significantly. Considering the combined effect of topographic factors on vegetation, significant changes in NDVI occurred at elevations lower than 5000 m and slopes of 0°–25°. These results reveal the complex relationship between topographic factors and vegetation dynamics, and provide valuable insights and management strategies for ecosystem responses in different subregions. This will be helpful for pasture conservation and alpine environmental management policies, which will ultimately promote the coordination of ecological restoration and economic development on the Tibetan Plateau.

**Author Contributions:** Y.L. conceived the original idea and designed the study. Y.L. processed and analyzed the data and wrote the manuscript. J.G. provided insights on the revision of the manuscript and suggestions for improvement. Y.Z. provided help for data processing and mapping. All authors have read and agreed to the published version of the manuscript.

**Funding:** This research was funded by the Second Tibetan Plateau Scientific Expedition and Research (Grant No. 2019QZKK0603).

**Data Availability Statement:** The data that support the findings of this study are available from the website given in the manuscript.

**Conflicts of Interest:** The authors declare no conflict of interest.

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
