# Peer review of "Investigating the Relationship between Topographic Factors and Vegetation Spatial Patterns in the Alpine Plateau: A Case Study in the Southwestern Tibetan Plateau"

_remotesensing, doi:10.3390/rs15225356_

Round 1
Reviewer 1 Report (Previous Reviewer 2)
Comments and Suggestions for Authors
I have reviewed this work twice before and rejected it due to some serious flaws. This is the third time I have read it and I acknowledge the work has improved but the improvement is not substantial enough, and it has not addressed the most obvious problems. I suggest a major revision due to problems in methodology, interpretation of results, and the writing style. I hope the authors can seriously consider the suggestions and help build a better remote sensing community by providing high quality manuscripts.
Usually, reviewers read a manuscript more carefully than future readers. If a reviewer cannot easily spot the merits of the manuscript, then a future reader will less likely find them. This is often due to bad writing habits, and honestly, it is often not even because of an English language problem, but because the manuscript lacks basic writing skills that can effectively and efficiently (concisely) convey ideas.
One minor example in the methods section is that the manuscript intentionally chose terms that are challenging to perceive on a first read. For example, the equation (1) is simply the slope of the liner regression, and every reader knows linear regression. Thus, by saying “linear regression” the manuscript will better help readers understand the equation, while using the term “long-time-series” may scare away potential readers. Another example is equation (2). The equation is very hard to understand, but it is equivalent to (Sie/Se)/(Si/S), which is a ratio between two probabilities. The Sie/Se is the conditional probability P(vegetation change| topography), and Si/S is the P(vegetation change). Comparing the above two probabilities is straightforward and easy to understand for many readers, while the “distribution index” can confuse readers. These are two examples of why choosing a better term can largely improve the readability of the manuscript, and more clearly present the ideas.
I will point out more writing issues, but before that I need to talk about the problems in the methodology and interpretation of results.
I disagree with using the word “impact” in the result section. I have stated this point in previous reviews. The results should present what has been studied by the authors using the abovementioned methods. The result section should only present facts (data) of what has been analyzed by the authors, usually not findings and conclusions conducted by other studied. Which methods were used? The NDVI trend by linear regression, the comparison of probabilities (distribution index). Therefore, it is appropriate to present which type of topography occurs with which type of NDVI change, as these have been studied by the authors. However, can the results say the topography impacts the vegetation? No, because impact means physical processes which were not studied by the methods. The result can even talk about correlations and regressions but cannot imply any impacts. I acknowledge there could be some impacts, and the authors are welcome to discuss the impacts in the discussion section, but not in the result section.
There are many examples of bad writing. As a reviewer I cannot point out every instance, so I will show some examples and a future version can address them. One obvious example is the paragraph introducing the metric P.
L225-227: I can guess this is about some conditional probability, but why let the reader guess? Why not write it more clearly?
L227-228: The however sentence is useless unless it is specific. It sounds like something is more important than something else. But one can find such examples almost everywhere on anything.
L228-230: What phenomenon? What is irrationality and uncertainty?
L230-232: Where was this distribution index mentioned before? The reader has no clue what it is.
L232-235: The index may be useful, but without knowing its definition, the entire sentence is useless to the reader.
L235: This is the last chance to say the definition of the distribution index, but the authors failed to say, and let the reader guess what it is.
So, instead of confusing the reader, the authors can simply give the CORRECT definition in the beginning of this section, explain the variables, and briefly interpret some example values.
Line specific comments
L48-49: Higher NDVI simply means more (denser) green vegetation, right? How can that be better? For example, invasive species may have higher NDVI, but are they good? What does “a lower more inadequate” mean?
L50-70: This entire paragraph reads like chaos. The beginning talks about NDVI, then it talks about other factors. If there is too much information and too many points to make, why not split it into two paragraphs?
L53-55: need references
L58: Geographic is a pointless word here. Geographic environments can be interpreted as different locations. So, the sentence reads like “vegetation is different in different locations”. But isn’t it obvious? I think geomorphic or topographic makes more sense, but not sure 15 and 16 can support this point.
L71-73: The first sentence is not related to the bulk of the paragraph. This paragraph describes general scenarios. Leave the TP for the next paragraph, which describes the study site.
L133-147: This paragraph is another example of bad writing. Firstly, it talks about “our objective is to…” (L135) and “we aim to…” (L136). I thought these are the objectives of this study. Then L138-124 lists three objective. Finally, “this study aims to …” (L142) shows again and I am totally confused what the objective of this study is. Further, the three numbered objectives are different to read and need editing.
L147: The acronym was never used in the manuscript, so why make it? It only adds more burden to the reader as the reader must remember the acronyms. I guess this is a more political term and it may help in a grant proposal to secure funding, but for an international scientific journal, this acronym is useless. This sentence can simply be “improve regional ecological management and sustainable development”.
L246-248: What purpose have these two opening sentense achieved? In my opinion, repeating the figure caption in the opening sentence does more harm than good. The opening sentence is the single most important sentence in a paragraph. Ideally, it should be the take home message and conveys the most important idea of what the authors want to say. It should be very imformative, and figure names, if necessary, can be put in parentheses. A reader may not have interest and time to read the entire paragraph, but may read the opening sentence of each paragraph.
L249: Is it -0.030 or 0.030? This obvious mistake should not be in a submitted manuscript for review. Remember, reviewers are not supposed to be editing the manuscript. Only polished manuscripts should be submitted.
L249: What does most mean? How much is most? Why not use a number?
L255-258: This is a possible explanation of the results, and should not be here. It belongs to the discussion.
L258-259: The ecological quality? Is this what the result is about? The result here should focus on NDVI and NDVI changes. This type of interpretation must not be in the result section.
L262-263: Interpretation again. Move to discussion.
L264-266: This is the opening sentence I was looking for. Polish it and move it upward.
Sections 3.2 and 3.3: I think there is some overlapping between the two sections. Some information contained in Figs 8-10 were already mentioned in Figs 5-7, is it correct? If so, consider merging those sections and make the manuscript more concise and readable.
L273: What does “which” refer to? Unclear here.
L284-286: The sentence lacks the subject.
Figs 5-7: The captions said NDVI changes but the legends said NDVI. I am confused.
Fig 8: Defind the long term “changes of in NDVI variation types”.
Section 4.1: I suggest moving all the “impact” words here. For example, the authors can link their findings to other studies which have examined the physical processes of topography influences on vegetation / vegetation changes. Educated guesses and explanations can be made. What has been written here reads pointless, and the explanations are too generalized. I would like to see more insights if the authors prefer keeping the “impact” words in the discussion section.
L511: “May be due to forests mainly occuring below …” I am surprised to see this vague guessing in a remote sensing study. The global land cover maps are available at much higher resolutions and the authors have access to such data. I would like to see more solid findings and explanations.
L515-516: Need references to support the point that vegetation is more fragile. Again, by adding references, it creates the opportunity for the authors to build connections between this work and other studies, and helps write a better discussion section.
L524-541: The discussion in this paragraph is generally superficial and needs to be more specific and more in-depth. The authors also agreed to “deeply explore…” on L537.
Comments on the Quality of English LanguageSee the comments above.
Author Response
Please see the attachment.

Reviewer 2 Report (New Reviewer)
Comments and Suggestions for Authors
The paper has conducted a thorough analysis and discussion, and it is recommended to supplement the research background content to enrich the research foundation.
Comments on the Quality of English LanguageModerate editing of English language required
Author Response
Please see the attachment.

Reviewer 3 Report (New Reviewer)
Comments and Suggestions for Authors
The manuscript utilized NDVI data and topographic data to analyze the distribution of NDVI in various single and combined topographic conditions in the Southwestern Tibetan Plateau (SWTP) over the past two decades. The study is interesting and the insights gained from this study can contribute to future ecological conservation and restoration of the SWTP. The results of the paper are analyzed deeply and thoroughly. However, there are still several issues that require further modification or clarification.
1.Two kinds of data (MODIS-based NDVI data, ASTER GDEM elevation data) were used in the research, and previous methods (Trend analysis and Distribution index) were directly used to carry out the research. Where is the innovation of this paper, need to further highlight the expression.
2.As the authors said, this work divided elevation, slope, and aspect into five grades. This scale may still be relatively coarse for analysis,especially for altitude(the interval of 500 meters is relatively large) and aspect, which could potentially causing analytical errors. Why does the authors not increase the level (say, to 7 levels or 8 levels) according to the local reality to depict the impact of terrain features on NDVI in more detail?
3. The description of the method is lacking in some significant respects. It is recommended to add relevant content. Including the selection of Trend analysis and Distribution index, is there any other methods can be used?
4. English should be checked and improved.
Comments on the Quality of English LanguageCan be improved.
Round 2
Reviewer 1 Report (Previous Reviewer 2)
Comments and Suggestions for Authors
The authors have made a great effort in revising the manuscript. The presentation is much more clearer than the previous versions. The manuscript now reads smoothly and should be a good contribution to the remote sensing community.
Reviewer 3 Report (New Reviewer)
Comments and Suggestions for Authors
Alll the question and comments were addressed well.
This manuscript is a resubmission of an earlier submission. The following is a list of the peer review reports and author responses from that submission.
Round 1
Reviewer 1 Report
Comments and Suggestions for Authors
The issue of the paper is important. It has covered a complex geographical region. It is interesting, especially from the methodological point of view. The comments are given below.
Minor comments
The paper claims that the Tibetan Plateau has a typical Mountainous ecosystem. It would be better to explain this situation a little bit more so that the outside community understands the situation. In addition to the description of the topography in general, rainfall and temperature situation, elevation ranges, and types of topography, major human activities might be discussed in brief.
The literature cited (line 48) mentioned that topographic factors- light, temperature, water, and soil nutrients- exert different effects in a certain area. If these are the major factors, how far these factors are considered in this study?
It has mentioned that 'a series of environmentally friendly policies such as reverting cultivated land to grassland, afforestation, natural forest resource protection, and grassland ecological protection projects, etc.' have been implemented and have positively contributed too. How to assess the role/contribution of these policy interventions? It might be discussed the situation before these policies and the situation after the implementation of these policies.
It has mentioned the situation of overgrazing. It would be interesting to show the situation of livestock grazing density of the two periods, especially concerning significantly NDVI increasing and decreasing areas as shown in Figure 3b.
Line 228-231 claims that ' the 4500-5000 m range is the most critical distribution area of the vegetation, accounting for 52.33% of the significant decrease areas. The no significant decreases area primarily distributed in the 4500-5500 m section, whereas the 4500-5000 m range is the main area with no significant decreases of the NDVI.'. This explanation NEEDs to revisit as it has some confusion that 4500-5000 elevation zone mentioned significantly decrease area and also mentioned 4500-5000 as not significantly decreased area.
Major comments
The major concern is why there are changes in NDVI over time as the mentioned topographic factors such as elevation, slope and aspects are same over the time. Importantly, it has mentioned no major effect of slope and aspects.
It is, therefore, human intervention and the policy might have major impacts on NDVI change. It is expected to reflect on these two aspects. Furthermore, what changes have been observed in rainfall and soil moisture situations that might have major effects on vegetation?
Author Response
Q & C-1. The paper claims that the Tibetan Plateau has a typical Mountainous ecosystem. It would be better to explain this situation a little bit more so that the outside community understands the situation. In addition to the description of the topography in general, rainfall and temperature situation, elevation ranges, and types of topography, major human activities might be discussed in brief.
A: Thank you very much for your hard work and nice suggestions.
Thanks to your suggestion. More information on Tibetan Plateau had been added in the Introduction and Material and methods. Pleased to find them in the revised manuscript, especially in the fourth paragraph of the introduction and the 2.1 section of the study area.
Q & C-2. The literature cited (line 48) mentioned that topographic factors- light, temperature, water, and soil nutrients- exert different effects in a certain area. If these are the major factors, how far these factors are considered in this study?
A: Thank you very much for your suggestion.
As it mentioned in our manuscript, the environmental variability controlling the spatial patterns of land cover might be attributed to topographical factors and their influence on soil-water distribution. Lots of studies have demonstrated the influence of topography on the spatial patterns of vegetation [1]. A) In order to adapt to this environment for better survival and growth, vegetation will actively adapt to the environmental changes brought about by the elevation via the changes of leaf thickness, stomatal density, and leaf area. Therefore, elevation is the key factor affecting vegetation change, which indirectly affects vegetation types and spatial distribution by controlling the distribution of temperature and precipitation [2]. b) Slope can reflect the degree of slope of the terrain surface, which affects the surface's energy conversion and material flow. When the slope is large, the terrain is steeper, causing a loss of soil water and nutrients. This does not facilitate the rooting of vegetation in the soil and makes it more susceptible to natural disasters such as erosion, making vegetation unfavorable in areas with large slopes [3]. c) Aspect is the pointing or facing of the surface slope, which affects the distribution of solar radiation energy and soil moisture at the surface. The results show that sunny slopes receive more solar radiation than shady slopes, but this suppresses photosynthetic rates and accelerates evapotranspiration from the surface, which can result in relatively better vegetation cover on shady slopes [4].
Therefore, we reveal the influence of topographic factors on NDVI from three perspectives of elevation, slope, and aspect to provide a theoretical basis for local vegetation restoration and ecological environmental protection. Although this study can reveal the relationship between topographic factors and vegetation change types. Mechanistically, it can be said that the relationship between hydrothermal conditions, light, solar radiation, and soil nutrients on the growth status of vegetation and future change trends. However, how to quantitatively reveal the magnitude of each factor's influence on the NDVI is needed to be addressed in future studies.
Pleased to find the details from the revised manuscript.
References:
- Gomez-Plaza, A.; Martinez-Mena, M.; Albaladejo, J.; Castillo, V.M. Factors regulating spatial distribution of soil water content in small semiarid catchments. Journal of Hydrology 2001, 253, 211-226, doi:10.1016/s0022-1694(01)00483-8.
- Liu, C.L.; Li, W.L.; Wang, W.Y.; Zhou, H.K.; Liang, T.G.; Hou, F.J.; Xu, J.; Xue, P.F. Quantitative spatial analysis of vegetation dynamics and potential driving factors in a typical alpine region on the northeastern Tibetan Plateau using the Google Earth Engine. Catena 2021, 206, doi:10.1016/j.catena.2021.105500.
- Mendez-Toribio, M.; Meave, J.A.; Zermeno-Hernandez, I.; Ibarra-Manriquez, G. Effects of slope aspect and topographic position on environmental variables, disturbance regime and tree community attributes in a seasonal tropical dry forest. Journal of Vegetation Science 2016, 27, 1094-1103, doi:10.1111/jvs.12455.
- Burt, T.P.; Butcher, D.P. Topographic controls of soil-moisture distributions Journal of Soil Science 1985, 36, 469-486, doi:10.1111/j.1365-2389.1985.tb00351.x.
Q & C-3. It has mentioned that 'a series of environmentally friendly policies such as reverting cultivated land to grassland, afforestation, natural forest resource protection, and grassland ecological protection projects, etc.' have been implemented and have positively contributed too. How to assess the role/contribution of these policy interventions? It might be discussed the situation before these policies and the situation after the implementation of these policies.
A: Thank you very much for your suggestion.
There are several environmental policies had been implemented and resulted in vegetation greening and ecological improvement, especially in the Tibetan Plateau. However, this paper focuses more on the influence of topographic factors change on vegetation (NDVI). So, we only discussed a little about the ecological changes in the SWTP before and after the implementation of these policies. However, as the reviewer said that topographical factors could limit human activities and affect where and how the policies are implemented. Therefore, we mentioned in the discussion section that we need to adapt to local conditions. It is urgent to adopt different conservation strategies to restore and build the ecological system of the SWTP, to support the sustainable development of alpine pasture-livestock system.
In our previous study, we revealed the changes in vegetation NDVI over the last 20 years and the relationship between vegetation cover and grazing activities in the first and second decades of 21st century [1]. We found that with the implementation of the Grassland Ecological Protection Program (2011-2020), the livestock load was significantly reduced, and the vegetation was significantly improved in the second decade than in the first decade. However, it is very difficult to quantitatively analyze the effect of each ecological restoration project/policy due to the variety of the implementation duration and areas. Therefore, the ecological impacts of the implementation of these policies were not revealed in a direct way. More efforts are still needed to explore the impact of ecological projects on the NDVI for the ecological conservation in different time and spatial scales, to provide a scientific basis for environmental protection and human well-being in the future.
Pleased to find the details from the revised manuscript.
References:
- Li, Y.; Gong, J.; Zhang, Y.; Gao, B. NDVI-Based Greening of Alpine Steppe and Its Relationships with Climatic Change and Grazing Intensity in the Southwestern Tibetan Plateau. Land 2022, 11, doi:10.3390/land11070975.
Q & C-4. It has mentioned the situation of overgrazing. It would be interesting to show the situation of livestock grazing density of the two periods, especially concerning significantly NDVI increasing and decreasing areas as shown in Figure4b.
A: Thank you very much for your suggestion.
The main purpose of Figure 4 is to reveal the distribution pattern and change trend of vegetation in the SWTP, and to pave the way for further exploration of the spatial distribution relationship between vegetation change characteristics and topography. The results in Figure 4b show a significant decrease of NDVI in Zongba county, which may be because the southern part of Zongba county is the Himalayan range and the central part contains the Gangdis Mountains. But the vegetation is damaged in the relatively gentle terrain area between the two mountain ranges. We consider that this may be due to the fact that the region belongs to the valley of the Yarlung Zangbo River, with open topography and good vegetation growth in the region, which is mainly dominated by grazing behavior, thus leading to the significant decline characteristic of Zongba County in Figure 4b.
Pleased to find the details from the revised manuscript.
Q & C-5. Line 228-231 claims that ' the 4500-5000 m range is the most critical distribution area of the vegetation, accounting for 52.33% of the significant decrease areas. The no significant decreases area primarily distributed in the 4500-5500 m section, whereas the 4500-5000 m range is the main area with no significant decreases of the NDVI. This explanation needs to revisit as it has some confusion that 4500-5000 elevation zone mentioned significantly decrease area and also mentioned 4500-5000 as not significantly decreased area.
A: Thank you very much for your suggestion.
In this section, instead of using the traditional calculation methods, like the area of a change type on an elevation gradient/total area of the study area, we use the area of a change type on an elevation gradient/total area of that change type. This result can illustrate the variation of the type of change over five gradients. For example, the vegetation showed a significant decrease in the percentage of area on the five elevation gradients in the following order: 3.77%, 11.44%, 47.66%, 31.65% and 5.48%. It is mainly concentrated in the areas of 4500-5000m (47.66%) and 5000-5500m (31.65%), accounting for 79.31% of the total significantly reduced area. The manuscript had been revised to remove the misunderstanding of the description.
Pleased to find the details from the revised manuscript.
Major comments
Q & C-6. The major concern is why there are changes in NDVI over time as the mentioned topographic factors such as elevation, slope and aspects are same over the time. Importantly, it has mentioned no major effect of slope and aspects. Therefore, human intervention and the policy might have major impacts on NDVI change. It is expected to reflect on these two aspects.
A: Thank you very much for your suggestion.
We conducted a new analysis of the results of this study, which found that 0-15° was the area of dramatic vegetation change, with an increasing trend of vegetation on the shaded slopes and a decreasing trend of vegetation on the sunny slopes. More information had been added in the revised manuscript.
In addition, topography greatly limits human activities and implementation of the policy. Topography always influenced the way and direction of land use changes and has a strong spatial restriction on land resource allocation activities [1]. For example, due to the poor transportation and the ecological environment are very poor in the Ngari Prefecture, the desert grassland is the main type of vegetation in the area and one of the primary economic sources for the local population. Vegetation degradation is caused mostly by grazing activities in the SWTP [2]. Ecological problems such as pasture degradation and land desertification are becoming increasingly prominent. Despite the government's policy of grass compensation, grazing is the main source of economy, and the reduction of livestock numbers has led to great pressure on local production life and socio-economic development [3].
For policy and human activity related content, we have added in the discussion section.
Pleased to find the details from the revised manuscript with track changes, especially in 4.1 Influence of single topographic factor on the NDVI.
References:
- Li, Y.; Gong, J.; Zhang, Y.; Gao, B. NDVI-Based Greening of Alpine Steppe and Its Relationships with Climatic Change and Grazing Intensity in the Southwestern Tibetan Plateau. Land 2022, 11, doi:10.3390/land11070975.
- Chen, L.; Yang, S.; feng, X. Land use change characteristic along the terrain gradient and the spatial expanding analysis: a case study of Haidian district and Yanqing county, Beijing. Geographical Research 2008, 1225-1234.
- Li, Y.; Gong, J.; Dai, R.; Jin, T. Spatio-temporal Variation of Vegetation Cover and Its Relationship with Climatic Factors and Human Activities in the Southwest Tibetan Plateau. Scientia Geographica Sinica 2022, 42, 761-771, doi:10.13249/j.cnki.sgs.2022.05.002.
Q & C-7. Furthermore, what changes have been observed in rainfall and soil moisture situations that might have major effects on vegetation?
A: Thank you very much for your suggestion.
We found that aspect affects vegetation, and the study results showed an increasing trend of vegetation on shady slopes and a decreasing trend of vegetation on the sunny slope. This may be because aspect is the main reason for controlling soil moisture in the subareas when elevation is not enough to cause vegetation changes. The sunny slopes always get more solar radiation than that of the shady slopes, which will suppress the photosynthetic rate of vegetation and accelerate transpiration at the surface, then resulted in a relatively better vegetation cover on the shady slopes in the alpine areas [1-2].
Pleased to find the details from the revised manuscript with track changes, especially in Line 485 to 492.
References:
- Burt, T.P.; Butcher, D.P. Topographic controls of soil-moisture distributions Journal of Soil Science 1985, 36, 469-486, doi:10.1111/j.1365-2389.1985.tb00351.x.
- Yirdaw, E.; Starr, M.; Negash, M.; Yimer, F. Influence of topographic aspect on floristic diversity, structure and treeline of afromontane cloud forests in the Bale Mountains, Ethiopia. Journal of Forestry Research 2015, 26, 919-931, doi:10.1007/s11676-015-0155-4.
Reviewer 2 Report
Comments and Suggestions for Authors
I am rejecting this manuscript because it has three serious problems. 1) The authors seemed to have done lots of analyses, but the study lacks insight and depth. 2) A potential academic integrity problem. 3) The methodology is also arbitrary in a few places.
The topographic control of vegetation is a well-known topic and is in every ecology textbook. After reading the manuscript, I am disappointed that I learned basically nothing new from it. If the authors attempt to study the “impact of topographic variables” (see L217, 218, 241, 255, 269, 274, 292, 303) or the “effect of topographic factors” (see L315, 349), then a physical model is needed (e.g., https://agupubs.onlinelibrary.wiley.com/doi/epdf/10.1029/2006WR005595) instead of linear regression (equation 1) and juxtapositions (Figs 4-11). The analyses presented here can only confirm that certain NDVIs and certain NDVI trends concur with certain landscapes, but do not explain the “impact” or “effect”.
As the authors pointed out, many factors contribute to NDVI distribution and NDVI change. The authors had just published another paper in the same study site and concluded that human activities and precipitation have significant influences on NDVI change in 2000-2020 (see https://www.mdpi.com/2073-445X/11/7/975, cited as [30] in this manuscript). If the authors already knew human activities and precipitation are important, why did they study the topography while ignoring those two factors? Or why didn’t they include topography in the published paper?
About the method, I am not convinced by the arbitrary classifications of elevation, slope, and aspect (Table 2). Why did the authors pick the 500 m interval in elevation, because 500 m looks better? The distribution index (equation 2) is sensitive to area change. If the study site shrinks or expands, some indices will certainly change. How does that affect the conclusion? At the very least, the authors need to conduct a sensitivity analysis of this index.
Line specific comments
L2: I suggest revising the title. I read it many times and still find “topographic distribution characteristics of NDVI” unclear.
L32-66: “Climate change” was mentioned six times in these two paragraphs, but the authors later clearly said they don’t study climate change (L471-472). Please revise the text as it is misleading to readers.
L92-100: move to site description
Fig 1: Blue is lake and green is river. Fix the legend.
L133: What is MOD13Q1? Do you assume every reader is an expert in MODIS?
L134: What is 13d?
L136: add a reference after MVC
Fig2: Add histograms
Equation1: This is the slope of linear regression, so it is trend not variation. Fix this problem on L166.
L174-175: Is this a conclusion of the analyses? Why does conclusion come before results?
Figs 10, 11: It’s very difficult to interpret the legend. What is the second Y axis? Why does the X axis repeat three times, and four times in the next figure? The authors should make them clear to readers.
L386-389: This echoes my first comment. The study adds little new to our existing knowledge in ecology.
L413, 418: If authors decide to talk about human activities, add a map and show those locations.
Author Response
Section 3. response to the Reviewer #2
Q & C-1. I am rejecting this manuscript because it has three serious problems. 1) The authors seemed to have done lots of analyses, but the study lacks insight and depth. 2) A potential academic integrity problem. 3) The methodology is also arbitrary in a few places.
A: Thank you very much for your suggestion.
We are sorry that this manuscript did not satisfy the reviewer. The manuscript had been revised according to your suggestion and comment, with a careful review and paper checking, especially on the three points you had mentioned.
Pleased to find the details from the revised manuscript, especially in the result analysis, methods and discussion.
Q & C-2. The topographic control of vegetation is a well-known topic and is in every ecology textbook. After reading the manuscript, I am disappointed that I learned basically nothing new from it. If the authors attempt to study the “impact of topographic variables” (see L217, 218, 241, 255, 269, 274, 292, 303) or the “effect of topographic factors” (see L315, 349), then a physical model is needed (e.g., https://agupubs.onlinelibrary.wiley.com/doi/epdf/10.1029/2006WR005595) instead of linear regression (equation 1) and juxtapositions (Figs 4-11). The analyses presented here can only confirm that certain NDVIs and certain NDVI trends concur with certain landscapes, but do not explain the “impact” or “effect”.
A: Thank you very much for your hard work and nice suggestions.
We have carefully read the article you had mentioned. After a careful study of the literature, we are not using a physical model, but a simple linear regression and a parallel equation for the calculation. The reasons are as follows:
- For the SWTP, which is a large area (308,900km2), there is no coupled soil-vegetation-hydrology model that can be better applied to a large scale. Therefore, this paper does not use physical models to explore the simple relationship between topographic factors and vegetation [1].
- Due to the terrain constraints resulting in local transportation difficulties and a poor ecological environment, these factors limit the installation of meteorological stations and observation sites. Various underlying data and observations are relatively scarce, and it is difficult to obtain observations of long time series and the input data required for the model. The data we can use in the SWTP is only from remote sensing data, which is an essential reason for limiting our use of physical models.
- In fact, simple regression analyses and juxtapositions are not necessarily wrong. As the reviewer suggested, our study used regression analysis methods to obtain NDVI trends and then calculated the dominance of NDVI trends under different topographic combinations with the help of distribution indices. Among them, regression analysis methods are commonly applied to determine the trend of long-time series data over time; topographic indices can indicate the distribution of different vegetation change types over different topographic positions. Both research methods have been used and validated in numerous academic studies [2-5].
In our research, the results of the study were able to confirm that NDVI and variability trends are related to the topographic combination. However, the influence of each topographic factor on vegetation is a complex and integrated process, and cannot be explained solely as a direct effect of topographic factors. The results of this paper can still illustrate the spatial constraint of topographic elements on NDVI and NDVI change trends in the SWTP. It can provide a clearer understanding of the influence pattern of topographic factors on vegetation distribution, so as to better grasp the current situation and future development trend of vegetation. And the results can provide policy makers with the ability to consider issues from the perspective of topographic factors and vegetation change trends when developing town development and ecological conservation projects. Therefore, we believe that the research method of this paper is logical and has some scientific significance.
At the same time, we think that the physical model mentioned by the reviewer is a good research direction. We can conduct a more in-depth study based on this study and use the results of this study as a typical region selection. For example, our study found that in the area of 4000-5500 m in elevation, 0-25° in slope, is the area with the most drastic vegetation changes. In the future, we can select observation sites in the range of 4000-5500 m and 0-25°. Then, we could use physical models to investigate the soil-vegetation-hydrology interconnection in this area and investigate the mechanistic link between ecological processes, hydrological processes, and slow soil processes. Perhaps the results of study can have a better explanation and extended meaning to the findings of this paper. These are our views on the issue, and we have added the corresponding content to the discussion.
Pleased to find the details from the revised manuscript with track changes, especially in Line 564 to 572.
References:
- Li, Z.K.; Li, X.Y.; Zhou, S.; Yang, X., F.; Fu, Y.S.; Miao, C.Y.; Wang, S.; Zhang, G.H.; Wu, X.C.; Yang, C.; et al. A comprehensive review on coupled processes and mechanisms of soil-vegetation-hydrology, and recent research advances. Science China Earth Sciences 2022, 52, 2105-2138.
- Li, H.D.; Jiang, J.; Chen, B.; Li, Y.K.; Xu, Y.Y.; Shen, W.S. Pattern of NDVI-based vegetation greening along an altitudinal gradient in the eastern Himalayas and its response to global warming. Environmental Monitoring and Assessment 2016, 188, doi:10.1007/s10661-016-5196-4.
- Liu, C.L.; Li, W.L.; Wang, W.Y.; Zhou, H.K.; Liang, T.G.; Hou, F.J.; Xu, J.; Xue, P.F. Quantitative spatial analysis of vegetation dynamics and potential driving factors in a typical alpine region on the northeastern Tibetan Plateau using the Google Earth Engine. Catena 2021, 206, doi:10.1016/j.catena.2021.105500
- Yu, H.; Zeng, H.; Jiang, Z. Study on Distribution Characteristics of Landsacpe Elements along the Terrain Gradient. Scientia Geographica Sinica 2001, 64-69.
- Chen, L.D.; Yang, S.; Feng, X.M. Land use change characteristic along the terrain gradient and the spatial expanding analysis: a case study of Haidian district and Yanqing county, Beijing. Geographical Research 2008, 1225-1234.
Q & C-3. As the authors pointed out, many factors contribute to NDVI distribution and NDVI change. The authors had just published another paper in the same study site and concluded that human activities and precipitation have significant influences on NDVI change in 2000-2020 (see https://www.mdpi.com/2073-445X/1/ 7/975, cited as [30] in this manuscript). If the authors already knew human activities and precipitation are important, why did they study the topography while ignoring those two factors? Or why didn’t they include topography in the published paper?
A: Thank you very much for your suggestion.
1) We note that climate warming and human activity have accelerated since the beginning of the Anthropocene. Long-term climate change and short-term human activities have led to ecological problems, such as glacial retreat, land desertification, and grassland degradation in local areas, which have significantly affected vegetation distribution patterns[1,2]. At present, climate change and the interrelationship between human activities and vegetation have become one of the main elements of the current global change in the research that explores the dynamic change pattern of vegetation. Therefore, in our previous study, we only investigated the effects of climate change and human activities on the vegetation of the SWTP[3,4].
2) Due to elevation, slope, and aspect data are not long-time series data, they do not fit well into previous studies with previous research methods.
3) In previous studies, we found that the SWTP is greatly limited by topographic factors from the spatial distribution of NDVI, variation trends, the extent of human activities, and grazing behavior of herders[5]. Therefore, the distribution index method is chosen in this paper to understand the distribution relationship between vegetation change trends and topographic factors to better grasp vegetation's current and future trends. It is expected that the results of this study will help to formulate scientific and reasonable policies for the construction of ecological civilization and can provide a scientific basis for ecological and environmental management and sustainable development of animal husbandry in the SWTP.
Pleased to find the details from the revised manuscript.
References:
- Tian, H.Z.; Yang, T.B.; Liu, Q.P. Climate change and glacier area shrinkage in the Qilian mountains, China, from 1956 to 2010. Annals of Glaciology 2014, 55, 187-197, doi:10.3189/2014AoG66A045.
- Zhan, Q.Q.; Zhao, W.; Yang, M.J.; Xiong, D.H. A long-term record (1995-2019) of the dynamics of land desertification in the middle reaches of Yarlung Zangbo River basin derived from Landsat data. Geography and Sustainability 2021, 2, 12-21, doi:10.1016/j.geosus.2021.01.002.
- Ichii, K.; Kawabata, A.; Yamaguchi, Y. Global correlation analysis for NDVI and climatic variables and NDVI trends: 1982-1990. International Journal of Remote Sensing 2002, 23, 3873-3878, doi:10.1080/01431160110119416.
- Piao, S.L.; Cui, M.D.; Chen, A.P.; Wang, X.H.; Ciais, P.; Liu, J.; Tang, Y.H. Altitude and temperature dependence of change in the spring vegetation green-up date from 1982 to 2006 in the Qinghai-Xizang Plateau. Agricultural and Forest Meteorology 2011, 151, 1599-1608, doi:10.1016/j.agrformet.2011.06.016.
- Li, Y.; Gong, J.; Zhang, Y.X.; Gao, B.L. NDVI-based greening of alpine steppe and its relationships with climatic change and grazing intensity in the Southwestern Tibetan Plateau. Land 2022, 11, doi:10.3390/land11070975.
Q & C-4. About the method, I am not convinced by the arbitrary classifications of elevation, slope, and aspect (Table 2). Why did the authors pick the 500 m interval in elevation, because 500 m looks better? The distribution index (equation 2) is sensitive to area change. If the study site shrinks or expands, some indices will certainly change. How does that affect the conclusion? At the very least, the authors need to conduct a sensitivity analysis of this index.
A: Thank you very much for your suggestion.
We have published a paper on the relationship between NDVI and elevation gradient in Research of Soil and Water Conservation[1]. In this paper, we graded the elevation gradient at 100m intervals, and the results showed that the different vegetation showed a general decreasing trend with increasing elevation. However, the NDVI of different vegetation types showed a fluctuating decreasing trend of a slight increase in decreasing with elevation gradient, which might be caused by the small elevation interval of 100m. For example, Figure 1(a) shows the results from the article on Research of Soil and Water Conservation and it shows the NDVI curve with elevation and frequency distribution of the meadow. The results show that the meadow have a fluctuating decreasing trend with the elevation, and when the elevation is graded at 100m, the data lack stability and fluctuate more. By reviewing the related literature, we found that due to the complex terrain and the drastic elevation change in the plateau area, the data stability will be reduced if the elevation grading step is too small [2]. Therefore, an elevation gradient set at 500-1000 m is a common choice. Combining the results of our previous study with the comparison of related literature, we found that grading when the step length is 500 m can better reflect the elevation gradient nature of the SWTP. The figure 1(b) shows the results of this paper. Secondly, when we consider the influence of combined topographic factors, the grading, if too small, will lead to too many topographic combinations and will not facilitate us to discover the distribution pattern of vegetation change types under the topographic combinations. Consequently, we finally set the elevation gradient to 500m.
Fig.1(a) Vegetation index curves with elevation and frequency distribution of meadow
Fig.1(b) Distribution of NDVI changes and percentage of change trend area at different elevations
When exploring the influence of topographic factors on the spatial distribution of vegetation, we would like to reduce it to the frequency of vegetation change types occurring on different topographic combinations. But as the reviewers noted, the comparability of the frequency of this distribution is also influenced by two factors:
- The distribution of terrain in space, that is the effect of area differences between;
- Different vegetation change types have different area weights, making it impossible to compare distribution characteristics between different types.
However, the author who proposed this formula encountered the same problem. In order to reveal the spatial distribution characteristics of landscape components under different topographic conditions, they proposed the formula of distribution index. Her article mentions the formula as a dimensionless index that eliminates the magnitude effect due to area differences. Thus, we directly use this calculation method of distribution index in this article. Detailed information can be found in this literature[3].
In response to the reviewer who raised the issue, we considered that this might result from relatively little explanation in the methodology. Therefore, we have improved the methodology section. Pleased to find the details from the revised manuscript, especially in Line 183 to 196.
References:
- Li, Y.; Dai, R.; Zhang, Y.X.; Gong, J. Spatiotemporal variation of vegetation NDVI and its relationship with altitude gradient in Southwest Tibet Plateau. Research of Soil and Water 2022, 29, 215-222, doi:10.13869/j.cnki.rswc.20210926.001.
- Li, H.D.; Jiang, J.; Chen, B.; Li, Y.K.; Xu, Y.Y.; Shen, W.S. Pattern of NDVI-based vegetation greening along an altitudinal gradient in the eastern Himalayas and its response to global warming. Environmental Monitoring and Assessment 2016, 188, doi:10.1007/s10661-016-5196-4.
- Yu, H.; Zeng, H.; Jiang, Z. Study on Distribution Characteristics of Landsacpe Elements along the Terrain Gradient. Scientia Geographica Sinica 2001, 64-69.
Line specific comments
Q & C-5. L2: I suggest revising the title. I read it many times and still find “topographic distribution characteristics of NDVI” unclear.
A: Thank you very much for your suggestion.
We have changed the title to “Exploring the spatial pattern of NDVI in different topographic combinations based on the distribution index in the Southwestern Tibetan Plateau”.
Pleased to find the details from the revised manuscript, especially in Line 2-4.
Q & C-6. L32-66: “Climate change” was mentioned six times in these two paragraphs, but the authors later clearly said they don’t study climate change (L471-472). Please revise the text as it is misleading to readers.
A: Thank you very much for your suggestion.
We had reorganized and rewritten the introduction. Pleased to find the details from the revised manuscript, especially in Part1. Introduction.
Q & C-7. L92-100: move to site description.
A: Thank you very much for your suggestion.
We had reorganized and rewritten the introduction and part 2.1-Case study area. Pleased to find the details from the revised manuscript.
Q & C-8. Fig 1: Blue is lake and green is river. Fix the legend.
A: Thank you very much for your suggestion.
We have revised figure 1 and its legend. Pleased to find the details from the revised manuscript, especially in Line 165.
Figure 1. The study area of the SWTP.
Q & C-9. L133: What is MOD13Q1? Do you assume every reader is an expert in MODIS?
A: Thank you very much for your suggestion.
We have reorganized and rewritten the method. The sentences on the data information and methods had been revised.
The new sentence is: “The NDVI is considered one of the most effective indicators to characterize the regional and global vegetation growth status and ecological environment changes. Among them, the data quality of NDVI data from MODIS is higher, and they are able to reflect the vegetation changes in the TP. The NDVI data used in this study are the MODIS datasets (MOD13Q1 product) for 2000–2020, with a temporal resolution of 16days and a spatial resolution of 250 m.”
Pleased to find the details from the revised manuscript, especially in Line 169-174.
Q & C-10. L134: What is 13d?
A: Thank you very much for your suggestion.
We're sorry that we didn't notice this problem. “16d” means “16 days”. The sentences on the data information and methods had been revised. The new sentence is: “The NDVI data used in this study are the MODIS datasets (MOD13Q1 product) for 2000–2020, with a temporal resolution of 16days and a spatial resolution of 250 m.”
Pleased to find the details from the revised manuscript, especially in Line 172-174.
Q & C-11. L136: add a reference after MVC
A: Thank you very much for your suggestion.
We have added the relevant literature after “MVC” as suggested by the reviewers. Pleased to find the details from the revised manuscript, especially in Line 176.
Q & C-12. Fig2: Add histograms
A: Thank you very much for your suggestion.
We have replaced the table with a histogram (Figure 3). The Figure 3 is shown below. Pleased to find the details from the revised manuscript, especially in Line 199.
Figure 3. Classification and area of topographic factors in the study area.
Q & C-13. Equation1: This is the slope of linear regression, so it is trend not variation. Fix this problem on L166.
A: Thank you very much for your suggestion.
The sentence had been revised and rewritten accordingly. The new sentence is “where θslope can indicate the trend of vegetation dynamics during the study period. Plus and minus slopes indicate positive and negative trends in vegetation dynamics;”
Pleased to find the details from the revised manuscript with track changes, especially in Line 208-209.
Q & C-14. L174-175: Is this a conclusion of the analyses? Why does conclusion come before results?
A: Thank you very much for your suggestion.
We found a real problem with this sentence. We removed it and rewritten this part. Pleased to find the details from the revised manuscript with track changes.
Q & C-15. Figs 11, 12: It’s very difficult to interpret the legend. What is the second Y axis? Why does the X axis repeat three times, and four times in the next figure? The authors should make them clear to readers.
A: Thank you very much for your suggestion.
We have revised the illustrations in Figure 11 and Figure 12.
The new sentence is “ Note: The horizontal axis from left to right is the aspect of NDVI ≥ 0.2, 0.2 < NDVI < 0.4, NDVI ≥ 0.4. The right Y-axis represents the magnitude of the distribution index of NDVI under a certain terrain combination, and the magnitude of the color scale indicates its value. Similar values in a fixed elevation interval belong to the same color, indicating that vegetation grows similarly in these terrains. The greater the P-value, the greater the distribution advantage of the vegetation under the terrain combination. The lower the P-value, the less suitable the vegetation is for growth under the given conditions. When the color is white, it means that such a terrain combination does not exist.”
Pleased to find the details from the revised manuscript with track changes, especially in Line 394-401.
Q & C-16. L386-389: This echoes my first comment. The study adds little new to our existing knowledge in ecology.
A: Thank you very much for your suggestion.
We performed a new analysis of the results in the article based on the recommendations of previous experts. When we discuss this in terms of a single topographic factor, the results of our study show that NDVI on shady slopes is greater than that on sunny slopes, the increasing trend of vegetation is concentrated on shady slopes, and the decreasing trend is concentrated on sunny slopes. However, when we integrated the three topographic factors of elevation, slope and aspect, we found that the influence factor of elevation was larger. The influence of slope and aspect is relatively small and almost negligible. Therefore, we believe that elevation, slope, and aspect are certainly important factors affecting the distribution pattern of the mountain vegetation. Still, the three's contribution depends on the study's scale and object[1].
The above is the result of our re-analysis, which may be a new development compared to the previous study results. We have rewritten the result and discussion section.
References:
- Han, J.; Shen, Z.H.; Ying, L.X.; Li, G.X.; Chen, A.P. Early post-fire regeneration of a fire-prone subtropical mixed Yunnan pine forest in Southwest China: Effects of pre-fire vegetation, fire severity and topographic factors. Forest Ecology and Management 2015, 356, 31-40, doi:10.1016/j.foreco.2015.06.016.
Q & C-17. L413, 418: If authors decide to talk about human activities, add a map and show those locations.
A: Thank you very much for your suggestion.
We have added those locations in Figure 1. Pleased to find the details from the revised manuscript with track changes, especially in Line 165.
Round 2
Reviewer 1 Report
Comments and Suggestions for Authors I have gone through the comments addressed by the authors and also observed the manuscript. I believe that the manuscript is fine now. Please proceed further. Thanking you.Reviewer 2 Report
Comments and Suggestions for Authors
This is my second time reviewing this work. I appreciate that the authors substantially edited the manuscript and responded previous comments. I am suggesting a major revision because the biggest problem was still not addressed.
Every researcher knows that correlation is not causation. The word “effect” was repeatedly used throughout the manuscript, and it is a strong word meaning causation. I agree that many factors, including human activity and climate change, can trigger vegetation change. But before more detailed research and solid field observations, it is too early to conclude that topography affects the NDVI and NDVI change. Certain topographic factors, either single or combined, occur with certain NDVI and NDVI change trends; this is the only certain conclusion based on the current analyses. The authors are welcome to explore the correlation or association with better statistical tools, but that still does not answer the “effect” question.
There is a more dangerous problem assuming topography causes NDVI change. Topography is “relatively constant” (L87) and if topography affects NDVI change, the effect can be assumed continuous. Then, given the age of the modern topography, even if we count from the beginning of Anthropocene, why is the modern NDVI still confined in [-1, 1]? It is reasonable to assume that NDVI should be wildly beyond this range. Note that if human activity or climate change was substituted for topography, then this problem could be delicately avoided, because the authors could claim that neither human activity nor climate change is a constant factor. Nevertheless, this problem is related to the causation/effect problem, that certain intermediate physical processes were skipped before the arbitrary conclusion that topography affects vegetation change was claimed. It could be possible that recent vegetation life cycle contributed to the NDVI change, and it just occurred with certain topographic conditions. But since the present analyses did not go into such detail, it should avoid touching the causation topic.
The discussion has been edited but needs more serious work. My general suggestion is to avoid guessing or use other study sites to explain what’s happening in Tibetan Plateau, and only write sentences directly associated with the presented analyses. For example, L617, the study [5] did not analyze topography and cannot support this sentence. L617-619: this sentence may be true but it’s too general, and I don’t understand its purpose. Is it a generalization based on the analyses in this study? Or is it a general statement providing an explanation to the observation of this study? This is also the first time that “temperature” is mentioned in this manuscript. I am curious how the authors knew the temperature caused the NDVI change. Similar problem on the next sentence L619-623; how did the authors know the change in leaf thickness, stomatal density, and leaf area in Tibetan Plateau? The citation [65] was a study in Ethiopia, and how did the authors know that similar things are happening in two remote places? I am not arguing that those statements are wrong, honestly, I prefer to think they are correct, but this is academic writing and readers are expecting findings based on facts, not guessing. I am not going to point out every similar issue in the discussion; the authors should make the efforts and overhaul this part.
L698-702: First, the wordy term “due to the fact that” can be reduced to one word: because. Second, “elevation and slope are more densely populated” makes no sense to the readers.
L744: the vegetation change type is increasing. Did vegetation species increase or did NDVI increase?
L746: Why will the sunny slope be used as restoration function area? Is the recent NDVI trend unnatural and bad? I don’t see this proposal much relevant to the presented analyses.